# Insights into Melanoma Clinical Practice: A Perspective for Future Research

**DOI:** 10.3390/cancers15184631

**Published:** 2023-09-19

**Authors:** Giang T. Lam, Carmela Martini, Tiffany Brooks, Sarita Prabhakaran, Ashley M. Hopkins, Ben S.-Y. Ung, Jingying Tang, Maria C. Caruso, Robert D. Brooks, Ian R. D. Johnson, Alexandra Sorvina, Shane M. Hickey, Litsa Karageorgos, Sonja Klebe, John J. O’Leary, Douglas A. Brooks, Jessica M. Logan

**Affiliations:** 1Clinical and Health Sciences, University of South Australia, North Terrace, Adelaide, SA 5000, Australia; 2Adelaide Medical School, University of Adelaide, North Terrace, Adelaide, SA 5000, Australia; 3Aware Women’s Health Private Clinic, Adelaide, SA 5006, Australia; 4College of Medicine and Public Health, Flinders University, Adelaide, SA 5042, Australia; 5Department of Surgical Pathology, SA Pathology at Flinders Medical Centre, Adelaide, SA 5042, Australia; 6Department of Histopathology, Trinity College Dublin, D02 PN40 Dublin, Ireland

**Keywords:** melanoma, survey, biomarker, clinical practice, diagnosis, prognosis, therapeutic options, patient management

## Abstract

**Simple Summary:**

This study acknowledges the challenges in melanoma diagnosis and the need for new technology that aids clinical decision making. Biomarkers that can accurately report on the underlying biology during melanoma progression are needed to enable an accurate diagnosis and prognostic risk stratification to provide new opportunities for personalized medicine.

**Abstract:**

Background: Early diagnosis is the key to improving outcomes for patients with melanoma, and this requires a standardized histological assessment approach. The objective of this survey was to understand the challenges faced by clinicians when assessing melanoma cases, and to provide a perspective for future studies. Methods: Between April 2022 and February 2023, national and international dermatologists, pathologists, general practitioners, and laboratory managers were invited to participate in a six-question online survey. The data from the survey were assessed using descriptive statistics and qualitative responses. Results: A total of 54 responses were received, with a 51.4% (*n* = 28) full completion rate. Of the respondents, 96.4% reported ambiguity in their monthly melanoma diagnosis, and 82.1% routinely requested immunohistochemistry (IHC) testing to confirm diagnosis. SOX10 was the most frequently requested marker, and most respondents preferred multiple markers over a single marker. Diagnostic and prognostic tests, as well as therapeutic options and patient management, were all identified as important areas for future research. Conclusions: The respondents indicated that the use of multiple IHC markers is essential to facilitate diagnostic accuracy in melanoma assessment. Survey responses indicate there is an urgent need to develop new biomarkers for clinical decision making at multiple critical intervention points.

## 1. Introduction

Cutaneous melanoma arises from the malignant transformation of melanocytes and is the most aggressive form of skin cancer, with a high propensity for metastasis [1]. Globally, there were approximately 325,000 cases of melanoma and more than 57,000 melanoma-related deaths in 2020 [2]. The incidence of melanoma is steadily increasing, with the highest rates observed in Australia, New Zealand, Europe, and North America [3]. With the number of newly diagnosed melanoma cases projected to increase by more than 50% by 2040, melanoma presents a significant burden for health care systems [3].

The early accurate diagnosis of melanoma is crucial to improve clinical outcomes, ensuring that patients receive appropriate treatment before the cancer progresses. Despite efforts to improve diagnostic approaches, the misdiagnosis and improper staging of melanoma remain well-recognized issues in dermatopathology [4]. In Australia, general practitioners commonly provide a clinical diagnosis that relies on the visual inspection of pigmented lesions following the ABCDE (asymmetry, border irregularity, color variation, large diameter, and evolution) rule [5,6]. However, it should be acknowledged that these features may not be consistent for all melanoma cases [7,8]. Due to the lack of melanin pigment and the notorious challenge in recognizing vascular patterns, amelanotic or hypopigmented melanomas are prone to misdiagnosis or delayed diagnosis at an advanced stage [9,10,11]. Some melanomas can appear as small lesions (less than 6 mm in diameter), despite being invasive [12]. Moreover, the ubiquity of potential melanoma precursor nevi may place additional limits on the effectiveness of the visual approach for melanoma detection [13]. Thorough histology assessment is required to confirm melanoma diagnosis, but significant diagnostic hurdles persist [7,14].

While routine Hematoxylin and Eosin (H&E) staining is standard for melanoma diagnosis, there is intra- and interobserver subjectivity for the interpretation of histological features in melanoma and ambiguous lesions [7,15,16]. Indeed, the histological criteria favoring a melanoma diagnosis are not pathognomonic for melanoma and may occasionally be observed in a wide range of other benign skin pathologies (e.g., dysplastic nevi, blue nevi, Spitz nevi, Reed nevi, and lentiginous junctional nevi) [7]. Furthermore, there are borderline lesions with equivocal malignant potential, making a definitive melanoma diagnosis challenging [7]. Ancillary techniques, such as fluorescence in situ hybridization (FISH), comparative genomic hybridization (CGH), next-generation sequencing (NGS), and immunohistochemistry (IHC) with the commercially available markers (e.g., S100, human melanoma black (HMB)-45, Melan-A, microphthalmia transcription factor (MITF), and SRY-related HMG-box gene 10 (SOX10)), can be used to assist the histopathological diagnosis, but at present, there is no consensus on a standardized protocol for widespread clinical adoption [7]. It is important to understand the reasons for the lack of standardization and the unmet need in melanoma clinical practice, to devise a better approach for the development of future ancillary tests.

This single-point cohort study had a combination of qualitative and quantitative questions (i) highlighting the current difficulties faced by clinical teams in melanoma assessment, (ii) investigating the utility of available ancillary tools to enhance diagnostic precision, and (iii) identifying areas of interest from a clinical perspective for the focus of future research.

## 2. Materials and Methods

From April 2022 to February 2023, professionals who work in the fields of melanoma diagnosis and oncology-based medicine were invited to participate in a 10-min online survey using the Qualtrics XM platform (Qualtrics, Provo, UT, USA). Eligible participants included clinical pathologists, general practitioners, dermatologists, and laboratory staff. Various organizations in Australia, Ireland, and the USA were approached for access approval and assistance in distributing the survey to eligible participants, including the University of South Australia, SA Pathology, Peter MacCallum Cancer Centre, Dermpath Diagnostics, Trinity College Dublin, St. James Hospital Dublin, the Coombe Women and Infants University Hospital Dublin, Alfred Health, and the Melanoma Institute of Australia. The participants were invited by their organizations through membership emails, e-newsletters, or using personal Twitter/Instagram accounts. A recruitment flyer was also disseminated to local dermatology clinics in Adelaide, South Australia, to enhance the participation rate. This research project was approved by the Human Research Ethics Committee of the University of South Australia (Ethics ID 204510) as required by the Australian government, specified in the National Statement on Ethical Conduct in Human Research (2007–updated in 2018).

This single-point cohort study was designed with a combination of qualitative and quantitative questions (*n* = 6). The respondents were first asked to indicate the location of their workplaces. In the next question, the respondents were asked how many cases of melanoma they diagnosed in a typical month and what percentage of these melanoma diagnoses were initially ambiguous. The respondents were then asked to select which cell markers they use to confirm melanoma diagnosis, if any. The respondents subsequently used a tick-box format to indicate their opinions on which aspects they think should be addressed in future melanoma research and optionally share their experiences or comments about melanoma research.

The responses from Qualtrics XM were transferred and analyzed using independent Student’s t-tests (for continuous variables) and chi-square tests (for categorical variables), using GraphPad Prism version 9.4.1 (GraphPad Software Inc., La Jolla, CA, USA) and IBM SPSS statistics version 28.0.1.1 (14) (IBM Corp., New York, NY, USA). This study was powered for a minimum completion rate of 30%, as indicated within the institutional ethics approval. Due to the nature of the participant recruitment, the response rate could not be determined. In this paper, percentages of frequency counts are reported for respondent demographics and survey responses. Qualitative responses to the open-text questions were evaluated using a thematic content analysis method.

## 3. Results

There were 28 respondents who completed the survey, including general practitioners, as well as laboratory and specialist dermatology staff. A 51.8% completion rate of all mandatory questions from the 54 initial respondents was recorded. The majority of the respondents were from Australia (78.6%), 10.7% were from Ireland, and 3.6% of the respondents were each recorded from India, the Philippines, and the USA. No correlation was determined between the answers provided by the respondents and their geographical location. Many indicated they diagnose less than ten cases of melanoma in a typical month (64.3%), while 21.4% indicated between 11 and 19 cases per month (Figure 1). Only 10.7% of the respondents indicated they diagnose more than 20, and 3.6% of the respondents indicated they do not diagnose any melanoma cases in a typical month (Figure 1).

### 3.1. Melanoma Diagnosis

Most respondents (85.7%) indicated that less than 25% of melanoma cases diagnosed were initially ambiguous (Figure 2). Among the respondents, 7.1% reported to have 26–50% ambiguous cases among their monthly melanoma diagnoses, whereas only one respondent indicated that this number was more than 50% (Figure 2). One respondent indicated that none of the monthly melanoma diagnoses were initially ambiguous (Figure 2).

### 3.2. Ancillary Testing to Confirm Melanoma Diagnosis

The current use of IHC markers for ancillary testing to aid melanoma diagnosis was investigated using a multiple-choice question (these data are detailed in Table 1). The majority of the respondents (82.1%) used IHC to confirm a melanoma diagnosis, while 10.7% never requested an ancillary test, and 7.2% did not wish to say. The use of multiple markers was indicated in 69.6% of the responses, while 30.4% preferred a single marker.

SOX10 was the most requested marker (64.3%). S100 and HMB45 were equally requested (53.6%), followed by Melan-A (32.1%). The respondents also utilized a series of other cell markers, including PRAME, p16, Ki67, MITF, and a molecular panel (Table 1). Of those who requested multiple markers, 18.8% requested two markers, while 37.5% requested three or four markers. Most commonly, S100, HMB45, Melan-A, and SOX10 were selected together, followed by S100, and HMB45 with either SOX10 or Melan-A. Some respondents also selected a combination of S100, SOX10, and Melan-A, as well as S100 together with HMB45.

### 3.3. Future Considerations for Melanoma Research

Prognostic tests were perceived as the most important future direction (92.9%), followed by new diagnostic and therapeutic options (89.3%, Figure 3). Only 10.7% of the respondents did not perceive diagnostic tests, therapeutic options, and patient management as important future directions for research, while one participant did not consider prognosis a necessary focus for future studies (Figure 3). Furthermore, 3.6% and 7.1% of the respondents preferred not to indicate their thoughts on prognostic testing and patient management, respectively (Figure 3). Other suggestions for future melanoma research included predictive markers and tests, marginal and sentinel lymph node analyses, neoadjuvant therapy, next-generation sequencing, and disease pathogenesis.

More than one-third of the respondents (39.3%) provided optional feedback “…to share any additional experience in melanoma assessment or comments about melanoma research”. Interestingly, IHC marker-related responses comprised over 50% of the qualitative feedback (*n* = 6). More specifically, one respondent highlighted that single-marker use was deemed to be insufficient for melanoma diagnosis, necessitating a panel of IHC markers for better reliability. The need for new markers was also highlighted (*n* = 3). One respondent, who preferred the utility of three markers (S100, HMB45, and Melan-A), noted that “BRAF” and “PRAME” are also helpful. Another respondent, who preferred SOX10 as an ancillary test, stated that they use it for assessing intraepidermal melanocytic proliferations, or for determining the lineage of poorly differentiated tumors. Other feedback included one respondent who expressed concern over the lack of funding towards molecular testing for the diagnosis of ambiguous lesions. A variety of other important clinicopathological considerations when assessing melanoma were also raised, such as age, location of lesion, family/personal medical history, and the interaction between melanoma cells, antigen presenting cells, and innate immune surveillance.

## 4. Discussion

This study highlighted the current approaches and challenges faced by clinicians, where melanoma assessment and diagnosis are often subjective. Visual detection is the primary tool for the clinical diagnosis of melanoma; however, it relies on melanin distribution and vasculature patterns, which can be obscured [7,17,18]. The lack of melanin granules in amelanotic or hypopigmented melanoma may affect early detection using the color criterion within the ABCDE guidelines, and therefore may contribute to high rates of clinical misdiagnosis [9,11,19,20]. In a retrospective study at a general dermatology-based outpatient practice, up to a third of diagnosed melanomas were found to have a preoperative measured diameter of 6 mm or less, whereas tumor invasion was evident in more than two-thirds of these small-diameter cases [12]. Furthermore, the hidden location of the lesions and underestimation of early changes in appearance can also result in missed or delayed clinical melanoma diagnosis [21].

The histological assessment of H&E slides remains the gold standard to confirm melanoma diagnosis; however, the extensive morphological heterogeneity of melanoma with indistinct pathognomonic features sometimes results in a controversial interpretation of melanoma and other lesions, especially borderline lesions [7,22]. In a study involving a broad panel of American practicing pathologists, the diagnosis of samples spanning moderately dysplastic nevi to early-stage invasive melanoma was neither reproducible nor accurate [15]. Similarly, discordance in the routine histopathologic interpretation of melanoma and melanocytic pathologies has been reported in other retrospective studies [23,24], raising concerns regarding histological criteria that have long been accepted as the cornerstone for diagnosing melanocytic neoplasms. These challenges in melanoma assessment were reiterated in qualitative feedback, highlighting IHC markers as an adjunct to refining the histopathological diagnosis.

Although several ancillary techniques to enhance the accuracy of melanoma diagnosis exist, the practical considerations of cost, tissue requirement, and ease of use can affect their incorporation into clinical practice [25]. Current molecular tests (e.g., FISH, CGH, and NGS) require significant tumor tissue, rely on laboratory expertise [25], and are “not funded”, as pointed out by a respondent, ultimately hindering their wide adoption in clinical practice. In contrast, IHC is considered the most easily adopted test, as it overcomes the aforementioned restrictions and limitations [25]. While the rate of IHC utilization has significantly increased in recent years [26,27,28], properly validated and standardized markers for adoption in clinical melanoma practice remain an issue. In this study, more than 80% of the respondents currently requested IHC tests to confirm a melanoma diagnosis. The most frequent markers selected by the respondents included S100, SOX10, HMB45, and Melan-A, which is consistent with previous observations of markers currently used in melanoma diagnosis [27,29]. These markers are functional proteins for melanogenesis or melanocyte differentiation and exhibit a wide range of reported sensitivities and specificities, especially across the different subtypes of melanoma [7,30]. While these markers can be particularly useful to determine the melanocytic origin of a poorly differentiated lesion, they are not very informative on either the biological potential or the primary pathogenesis of melanoma [7,29,30]. The utility of PRAME in melanoma diagnosis was also recognized by some respondents (17.9%). Recent retrospective studies reported the clinical potential of this emerging melanoma marker in distinguishing benign from malignant melanocytic lesions, while also acknowledging its limitations in assessing certain melanoma subtypes [31,32,33]. Only a few respondents used proliferative and cellular markers (e.g., Ki67 and p16) to refine melanoma diagnosis. Although these markers can serve as indicators for tumor cell proliferation and growth, they are not specific for melanocytes and may provide prognostic rather than diagnostic value [7].

Interestingly, more than half of the respondents in this survey preferred multiple markers over a single marker. Indeed, one respondent stated that “no single immunohistochemical marker is able to distinguish benign from melanocytic malignancies. A panel of IHC markers is more useful”. This can be attributed to the fact that individual markers with less-than-optimal sensitivity and specificity have limited clinical utility and demonstrate a poor ability to differentiate melanoma from other melanocytic lesions [7,30]. As such, panels of markers have been suggested to achieve a higher level of reliability in melanoma diagnosis [34,35]. In this survey, the participants selected variable combinations of markers, confirming no standardized adoption across different clinical practices. Further investigation into the utility of multiple markers in melanoma diagnosis should be evaluated in a suitable clinical setting to determine if such advances could also overcome the diagnostic ambiguity reported here. Consequently, unless new reliable markers are developed, with superior specificity and sensitivity, it will not be feasible to establish a “one-size-fits-all” panel of markers to accurately diagnose such a highly heterogenous cancer as melanoma.

The current challenges in melanoma practice emphasize the necessity for new biomarkers to improve clinical decision making [25,36], and this was reiterated by the respondents’ feedback in this survey. Apart from the aforementioned diagnostic tests, most respondents in this survey also emphasized the significance of prognosis, therapeutic options, and patient management as important aspects for future melanoma studies. Furthermore, the discovery of new biomarkers that can report on the primary disease pathogenesis could enhance our understanding of melanoma biology to overcome these issues, guiding a precision medicine approach for patients [37,38]. Thus, newly identified biomarkers should be evaluated in the context of their biological relevance to the primary pathogenesis, while also considering the heterogeneity of the disease (accounting for tumor variation and clonal evolution) [7,25,39,40], before their potential clinical utility can be truly determined, as different subtypes of melanoma can exhibit distinct genotypes and immunophenotypes [30,41]. As suggested by a respondent in this study, “the interaction between melanoma cells, antigen presenting cells and innate immune surveillance is an important topic”. This holds great promise for understanding the mechanisms involved in tumor pathogenesis [42], and this may provide novel targets for biomarker development.

## 5. Conclusions

In conclusion, this cohort study acknowledges the challenges in melanoma diagnosis and the importance of IHC as an ancillary tool. The results highlight the shortcomings of currently available IHC markers and an urgent need to develop new reliable biomarkers to address clinical decision making. Biomarkers that can report on the malignant propensity of intermediate melanocytic lesions or inform on melanoma primary pathogenesis will not only enable an accurate diagnosis, but will also improve prognostic risk stratification to provide new opportunities for personalized medicine.

## Figures and Tables

**Figure 1 cancers-15-04631-f001:**
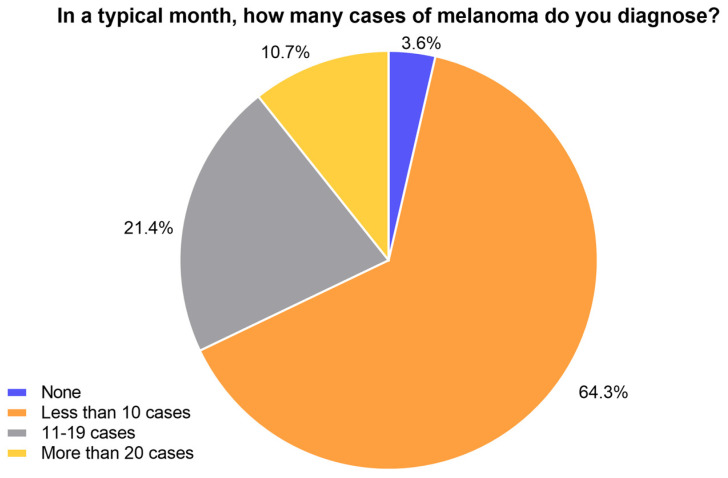
Melanoma diagnosis experience.

**Figure 2 cancers-15-04631-f002:**
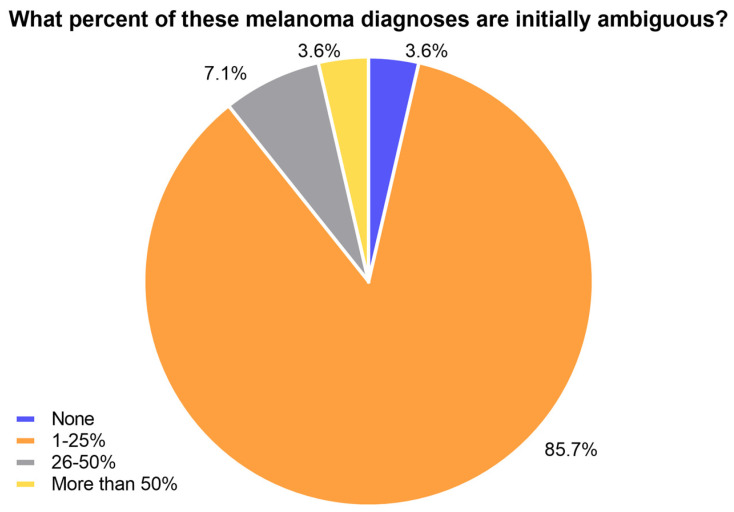
Ambiguity of suspicious melanoma lesions.

**Figure 3 cancers-15-04631-f003:**
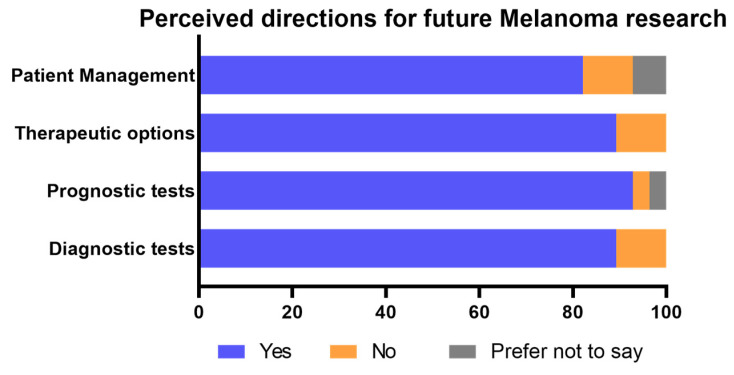
Opinions on important aspects for future melanoma research.

**Table 1 cancers-15-04631-t001:** Cell markers requested to confirm melanoma diagnosis.

**Cell Markers Used to Confirm a Melanoma Diagnosis**	**Frequency** (**%**)
SOX10	18 (64.3%)
S100	15 (53.6%)
HMB45	15 (53.6%)
Melan-A	9 (32.1%)
Never request ancillary diagnostic test	3 (10.7%)
Do not wish to say	1 (3.6%)
**Other Markers Used to Assist Diagnosis**	**Frequency** (**%**)
“PRAME”	5 (17.9%)
“p16”	2 (7.1%)
“ki67”	2 (7.1%)
“MITF”	1 (3.6%)
“Molecular Panel”	1 (3.6%)

PRAME, preferentially expressed antigen in melanoma; MITF, microphthalmia-associated transcription factor.

## Data Availability

Data are available for bona fide researchers who request them from the authors.

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
