# Peer review of "Insights into Melanoma Clinical Practice: A Perspective for Future Research"

_cancers, 2023, doi:10.3390/cancers15184631_

Round 1
Reviewer 1 Report
The manuscript titled "Insights into melanoma clinical practice: a perspective for future research" by Lam et al., addresses the challenges faced by clinicians in the diagnosis of melanoma. In particular the authors emphasize on the use of immunohistochemical (IHC) analysis for the presence (and relative abundance) of multiple markers as a reliable tool to be used in the diagnosis of melanoma.
The authors have evaluated the results of a survey using 54 responses with a 51.4%. completion rate. The authors conclude that SOX10 was the most frequently requested marker and most respondents preferred multiple markers over a single marker. The major caveat of this manuscript is that it does not constitute as a bonified research article, rather it is the analysis of the results of previously conducted survey. The validity of the survey is also hampered by the low response rate (slightly above half). .Rather than performing any actual experiments, the authors have relied on the results of a survey.
Moderate editing of English language required
Author Response
RESPONSE TO REVIEWER 1
The manuscript titled "Insights into melanoma clinical practice: a perspective for future research" by Lam et al., addresses the challenges faced by clinicians in the diagnosis of melanoma. In particular the authors emphasize on the use of immunohistochemical (IHC) analysis for the presence (and relative abundance) of multiple markers as a reliable tool to be used in the diagnosis of melanoma.
The authors have evaluated the results of a survey using 54 responses with a 51.4%. completion rate. The authors conclude that SOX10 was the most frequently requested marker and most respondents preferred multiple markers over a single marker. The major caveat of this manuscript is that it does not constitute as a bonified research article, rather it is the analysis of the results of previously conducted survey. The validity of the survey is also hampered by the low response rate (slightly above half). Rather than performing any actual experiments, the authors have relied on the results of a survey.
The authors acknowledge this manuscript is based upon the responses provided in a single-point cohort study with a combination of qualitative and quantitative questions. This study also supports previous primary biomarker research, and this has been further highlighted within the discussion. The authors have also highlighted no additional experiments were performed to test the use of the multiple markers and have referred to previous peer-reviewed publications within the literature.
The authors have included the following statement into the introduction:
LINE 62-63: This single-point cohort study with a combination of qualitative and quantitative questions(i) highlights […]
The authors have included the following statement into the discussion:
LINE 200-203: Further investigation into the utility of multiple markers in melanoma diagnosis should be evaluated in a suitable clinical setting to determine if such advances could also overcome the diagnostic ambiguity reported here.
The standard response rate for this type of clinical survey is around 30% as per our institutional ethics approval. The statistical power was also set at this threshold. At a result, the authors are confident in conducting the statistical analysis with the received response rate.
The authors have included the following statement into the materials and methods section:
LINE 87-88: The study was powered for a minimum completion rate of 30% as indicated within the institutional ethics approval.
Reviewer 2 Report
In the last paragraph of the introduction the authors should add what type of study this was and then follow with the aims.
Author Response
RESPONSE TO REVIEWER 2
In the last paragraph of the introduction the authors should add what type of study this was and then follow with the aims.
The authors acknowledge they should add what type of study this was before covering the aims. The study was designed as a single-point cohort study with a combination of qualitative and quantitative questions.
The authors have included the following statement into the introduction to clarify the study type:
LINE 62-63: This single-point cohort study with a combination of qualitative and quantitative questions(i) highlights […]
Reviewer 3 Report
Dear author
Greetings for your manuscript.
I have not revision to suggest.
Methods of survey are well described. Obteined data are well deteiled in figure and tables.
References are sufficient.
I hope we could replay a survey with more europea country.
Thank you
Author Response
The authors thank the reviewer for their kind feedback. We will discuss looking toward capturing data to address melanoma ambiguity in European countries as suggested
Reviewer 4 Report
In this study, the authors focus on the standardization of histological assessment for melanoma diagnosis, aiming to understand the challenges clinicians face during assessment and guide future research. They show that respondents favor using multiple markers over a single one and identified several areas for future research, including diagnostic tests, therapeutic options, and patient management. The conclusions stress an urgent need to develop new biomarkers to improve clinical decision-making in melanoma assessment. While the study is relatively simple, it still provides suggestive insights. One question that arises, however, is whether the authors have analyzed any correlation between the answers provided by the respondents and their geographical locations. Specifically, did the study uncover any differences in responses between urban and suburban areas, or between large and small cities?"
Author Response
RESPONSE TO REVIEWER 4
In this study, the authors focus on the standardization of histological assessment for melanoma diagnosis, aiming to understand the challenges clinicians face during assessment and guide future research. They show that respondents favor using multiple markers over a single one and identified several areas for future research, including diagnostic tests, therapeutic options, and patient management. The conclusions stress an urgent need to develop new biomarkers to improve clinical decision-making in melanoma assessment. While the study is relatively simple, it still provides suggestive insights.
One question that arises, however, is whether the authors have analyzed any correlation between the answers provided by the respondents and their geographical locations.
The authors tested for correlation between respondents and their geographical locations; however, no correlations were observed. The authors have reported this in the results section as per below:
LINE 97-98: No correlation was determined between the answers provided by the respondents and their geographical location.
Specifically, did the study uncover any differences in responses between urban and suburban areas, or between large and small cities?"
The authors did not request information regarding details of the respondent’s location apart from their country. This is an interesting suggestion, much of the demographic captured here is from Australia where these demographic differences are expected to be less pronounced; however, this may be something to consider for future cohort design particularly if the researcher was to focus on countries like America and Ireland.
Round 2
Reviewer 1 Report
The manuscript is suitable for publication.
Minor editing needed.